# Spatial Distribution Characteristics and Driving Factors for Traditional Villages in Areas of China Based on GWR Modeling and Geodetector: A Case Study of the Awa Mountain Area

**Shiying Li, Yuhong Song \*, Hua Xu, Yijiao Li and Shaokun Zhou**

College of Landscape Architecture and Horticulture Science, Southwest Forestry University, Kunming 650224, China

\* Correspondence: songyuhong@swfu.edu.cn; Tel.: +86-138-8858-2405

**Abstract:** Traditional villages are human treasures left behind by the integration of material space and non-material culture in the process of agricultural civilization. Studying the spatial autocorrelation characteristics, heterogeneity, and quantitative attribution of the factors influencing traditional villages provides new ideas for the protection of traditional villages. This study took 75 traditional villages as the research object. From the perspective of spatial autocorrelation and spatial heterogeneity, the study used nuclear density estimation and Moran's I index to analyze the spatial distribution patterns and selected 12 factors to construct the GWR modeling and geodetector to analyze the main driving forces and the interaction mechanism. The results showed that, firstly, the overall spatial layout of traditional villages in the Awa Mountain area had two cores, two sides, and a scattered distribution; the global Moran's I was 0.774, and 55.6% of traditional villages showed a clustering phenomenon. Second, the spatial layout of traditional villages in the Awa Mountain area has been jointly promoted and mutually constrained by multiple factors in a dynamic and complex mechanism with obvious spatial heterogeneity. The natural factor is the basic factor, which determines the location and scale of development of villages; the spatial factor is the auxiliary factor; the social factor is the decisive factor, with a negative global correlation and a positive local correlation; the regional cultural factor is the key factor, and the regional factor and the social factor complement each other; and factors such as a backward economic level, restricted transportation, less external communication, and low population density play a protective role. Third, the main driving factor is the proportion of ethnic minorities (X10), and the explanatory power of q-value reaches 0.54; the proportion of ethnic minorities (X10) ∩ average annual precipitation (X4) has the strongest interactive driving force, which belongs to nonlinear enhancement, and the q-value is 0.93, which proves that the explanatory power of the two-factor model is much greater than the single-factor model from the system perspective.

**Keywords:** Awa mountain area; traditional villages; GWR; geodetector

## 1. Introduction

A traditional village is a complex of human culture and natural environment in the process of human farming civilization [1]. The protection and continuity of traditional villages are of great significance to local cultural heritage, the construction of an ecological civilization, and the protection of cultural diversity. With the rapid development of modernization and industrialization, traditional villages in China are facing the problems of the loss of local characteristics, the weak local identity of residents, and the hollowing out of villages, and thus, the conservation of and research into traditional villages, is urgent [2]. The research on domestic, traditional villages has achieved rapid development in recent years, mainly focusing on sustainable development [3], planning and design [4], the cultural landscape [5], conservation and development [6], spatial morphology [7], and the spatial distribution characteristics [8]. Among these, studies of the spatial distribution

of traditional villages need to consider the influence of natural and human factors on the distribution characteristics of traditional villages, which can reveal the location preferences of traditional villages, the trend of spatial clustering, and the factors promoting and constraining the formation and development of villages, so as to promote the inheritance of vernacular culture and provide a basis for the formulation of strategies for protecting and developing traditional villages.

The spatial distribution characteristics of traditional villages are significantly related to factors such as the elevation, slope, slope aspect, population density, economy, distance from roads, and the distance from water sources [9]. Through spatial autocorrelation analysis, Huang Huijie et al. (2021) found that the spatial pattern of 314 traditional villages in the northwest region was influenced by a combination of natural and socio-human factors [10]; Jiao Jinying (2022) quantified the factors influencing the spatial distribution of 2424 traditional villages in the Henan section of the Yellow River Basin and revealed that natural factors such as rivers and the topography were the primary factors, followed by human factors [11]; Li Jining (2022) studied the spatial characteristics of 98 traditional village sites along the Silk Road of the Han Dynasty in the Ningxia region and found that central settlement and the location of transportation were the main influencing factors, and the influence of human factors was much greater than that of natural factors [12]. In summary, it can be seen that the influencing factors and formation mechanisms of the spatial layout characteristics of traditional villages are very complex, and the differences in different geographical spaces can lead to different research results. At present, the regions chosen for studies of the spatial distribution of villages have mostly focused on macro-territories, mainly in the national and provincial areas, and fewer studies have been from the perspective of minority gathering areas. The research methods used have mostly been based on ArcGIS spatial autocorrelation analysis, combined with residual analysis or correlation analysis; however, residual analysis cannot quantify the analysis factors [13], whereas correlation analysis can quantify but cannot express spatial heterogeneity [14]. The research has mostly analyzed factors such as the elevation, slope, slope aspect, precipitation, temperature, road network density, urbanization rate, and main industry [15–20], and less attention has been paid to spatial factors and regional cultural factors, etc. Therefore, this study tried to use GWR model and a geographic probe to study the less populated Awa Mountain area.

## 2. Study Area

The Wa Mountain Area (22–24° N, 99–100° W) (Figure 1) includes the Lincang and Pu'er regions, according to the Brief History of the Wa People (1986) [21–23]. The first to fifth batches of 75 traditional villages in total within 18 counties in the region published by the Chinese Traditional Villages Network (http://www.chuantongcunluo.com/) (accessed on 1 February 2023) [24] were selected as the target population. According to the results of the seventh census, the population of the Awa Mountain area is about 4,663,000, in which the population of ethnic minorities accounts for 49.9%, including Wa, Lahu, Yi, Dai, Brown, and other ethnic minorities living in the area. The terrain is complex, with very few flat terraces. It has a typical subtropical monsoon climate, rich biodiversity dominated by tropical plants, rich mineral resources, a maximum altitude of 3495 m, an average annual precipitation of 1108.7 mm, and an average annual temperature of 18.45 °C.

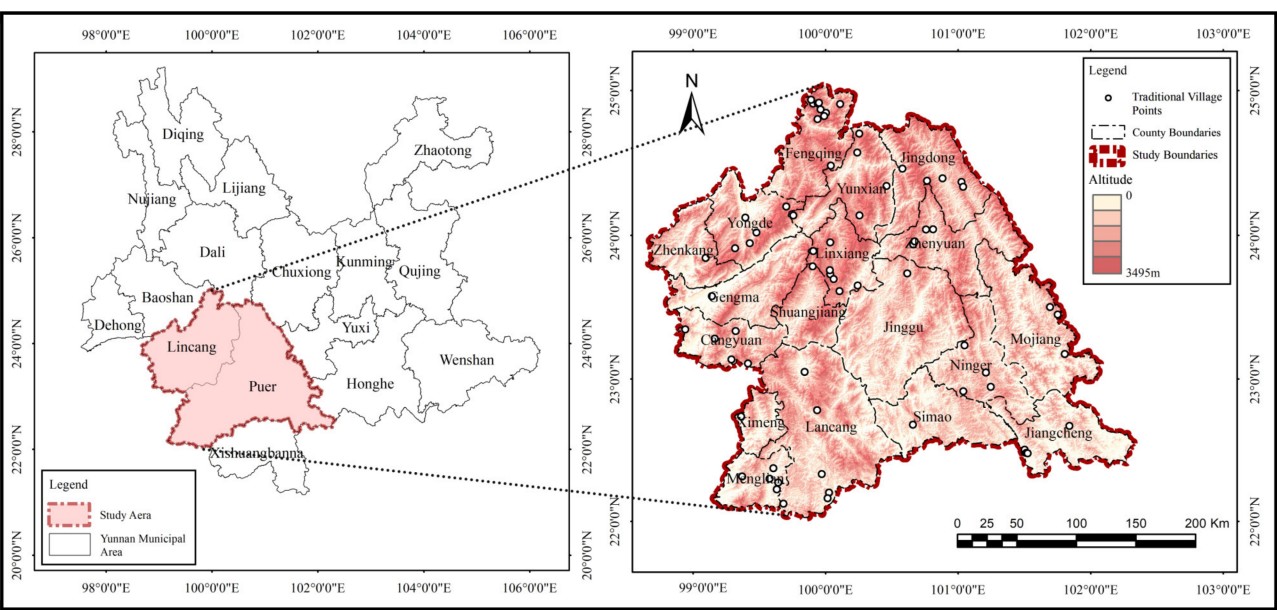

**Figure 1.** Location map of the study area.

## 3. Material and Methods

### 3.1. Data Sources

#### 3.1.1. Dependent Variable

The nuclear density mean (denoted as the dependent variable Y) can clearly reflect the distribution pattern of traditional villages in the study area. The latitude and longitude values of 75 traditional villages were selected with Gaode Software as the vector point database, and the administrative boundary vector data of the study area were obtained from the 1:4 million vector map information base provided by the National Basic Geographic Information Center (http://www.ngcc.cn/ngcc/) (accessed on 1 February 2023) [25]. Through a combination of previous experience and several experiments in ArcGIS10.2 (Esri, Redlands, CA, USA), the administrative boundaries of 18 counties were created with a network of 20,758 grids of 2 km × 2 km for analysis.

#### 3.1.2. Independent Variables

Most scholars' studies have considered the influence of natural, demographic, economic, and transportation factors. In order to build a more complete index system, this study incorporated regional cultural factors into the analytical model. The factors selected in this study were divided into four categories.

(1) Natural factors

(a) Topographic data: ASTGMT data were obtained from the geospatial data cloud (http://www.gscloud.cn/) (accessed on 1 February 2023) [26] through a mosaic, projection, and cropping to obtain DEM data of the Awa Mountain area with a 30 m resolution (the elevation was recorded as variable X1), and the slope and slope aspect information (recorded as variables X2 and X3) were obtained by using the 3D analysis tool in ArcGIS10.2. (b) Meteorological data. The raster dataset was obtained from the National Earth System Science Data Center (http://www.geodata.cn) (accessed on 1 February 2023) [27], and the annual precipitation and annual mean temperature were obtained using the raster calculator in ArcGIS10.2 (recorded as variables X4 and X5).

(2) Spatial factors

The distance of traditional villages from rivers and roads can reflect their spatial accessibility and convenience. Water sources are crucial to the layout of traditional villages, and are the main source of water for production and living, whereas the road network density is closely related to the terrain. Traditional villages with high road network density are more affected by modernization. Road and river data were obtained from the Open

Road Map (https://www.openstreetmap.org/) (accessed on 1 February 2023) [28], and the distances between the villages and these two features were calculated in ArcGIS10.2 using Euclidean distance (noted as variables X6 and X7).

(3) Social factors

The GDP can clearly reflect the level of economic development in the Awa Mountain area, as the local economy drives the development of villages. The total population is an important factor affecting the spatial distribution of traditional villages, which are created by people. The residents create the settlements, and the historical style and characteristics are changed with every move. GDP and total population data were obtained from the bulletin of the seventh population survey, the yearbook of Yunnan Provincial Bureau of Statistics (http://stats.yn.gov.cn/) (accessed on 1 February 2023) [29] (noted as variables X8 and X9).

(4) Regional cultural factors

The intangible cultural heritage is the existence of intangible traditional culture, and protected units of cultural relics are, to a certain extent, the tangible material carriers of intangible culture. The large population base of ethnic minorities in the study area is inseparable from the spatial distribution of traditional villages. The percentage of the minority population was obtained from the statistical bulletin of the national economic and social development of each state and city (recorded as variable X10). The data on intangible cultural heritage were obtained from China's Intangible Cultural Heritage Website (https://www.ihchina.cn/) (accessed on 1 February 2023) [30], and the data on protected cultural relics were obtained from the list of the first to eighth batches of the protected units of cultural relics published by the State Council (recorded as variables X11 and X12).

*3.2. Research Methodology*

3.2.1. Spatial Autocorrelation

(1) Kernel density estimation method

Kernel density analysis is usually used to calculate the density between each element and its neighboring elements, which can clearly reveal the distribution dynamics of the elements and has been widely applied in geographical studies [31]. Its expression is

$$f_n(x) = \frac{1}{nh} \Sigma_{i=1}^n k\left(\frac{x - x_i}{h}\right)$$

where $f$ is the traditional village density distribution function, $n$ is the traditional village distribution, $k\left(\frac{x-x_i}{h}\right)$ is the kernel function, and h is the set broadband.

(2) Global Moran's I

The global spatial autocorrelation and spatial distribution patterns can be clearly understood from the Moran's I index. Moran's I > 0 means a positive correlation, Moran's I = 0 means no correlation, and Moran's I < 0 means a negative correlation [32]. The spatial distribution patterns are classified as agglomerative, uniform or random distributions [15], for which the expressions are

$$I = \frac{n \times \sum_{i=1}^n W_{ij}(x_i - \overline{x})(x_j - \overline{x})}{\left(\Sigma_{i=1}^n \Sigma_{j=1}^n w_{ij}\right) \times \sum_{i=1}^n (x_i - \overline{x})^2}$$

where $I$ is the Moran's index value, $n$ is the number of county-level administrative regions in the Awa Mountain area, $x_i$ and $x_j$ are the observed values of county-level administrative regions, and $W$ is the spatial weight matrix expressing the proximity of n locations, where $W_{ij}$ is the proximity of regions $i$ and $j$.

(3) Local Moran's I

It is possible to estimate whether there is local spatial clustering of traditional villages with high and low values of distribution, classified as high–high (H–H), low–low (L–L), high–low (H–L), and low–high (L–H) clustering [33].

$$I_i = \frac{(x_i - \overline{x})}{m_0}\Sigma_j(x_i - \overline{x})$$

where $x_i$ is the number of traditional villages in unit $i$ of the study area and $\overline{x}$ is the average number of traditional villages.

### 3.2.2. Spatial Heterogeneity

(1) GWR model

Geographically weighted regression [34] (GWR) is used to study spatial heterogeneity according to the different degrees of influence of the independent variables on the dependent variables by assigning coefficient values. This study performed stepwise regression to analyze the influence of the factors in SPSS 22.0, compared the factors and selected the approach with the best fit before further GWR model regression. The regression equation is

$$y_i = \beta_0(u_i, v_i) + \beta_1(u_i, v_i)x_i + \beta_2(u_i, v_i)x_{2i} + + \beta_p(u_i, v_i)x_{pi} + \varepsilon_i$$

where $y_{i\_}i$ is the number of traditional villages in the $i$th county, $\beta_0$ is the intercept, $x_{pi}$ is the $p$th influential independent variable in the $i$th county, $(u_i, v_i)$ represents the geographic coordinates of the study area, $\beta_p(u_i, v_i)$ is the coefficient of the influencing factor in the $p$th county, and $\varepsilon_i$ is the random error.

(2) Geodetector

Geodetector [35] is used to reveal spatial heterogeneity and its drivers. In this study, it was mainly used to express the explanatory power as well as to detect the interaction between the independent variables from a system perspective *via* the expression:

$$q = 1 - \frac{\Sigma_{h=1}^{L}N_h\sigma_{h^2}}{N_{\sigma^2}}$$

where $L$ is the number of traditional villages, $N_h$, $\sigma_{h^2}$ is the number of cells and the variance of $h$, and $q$ denotes the explanatory power, which takes values between 0 and 1.

## 4. Results

### 4.1. Nuclear Density Analysis

In order to visually analyze the density distribution of traditional villages in the Awa Mountain area, based on several experiments and the relevant research of previous scholars [36], a 50 km bandwidth was used as the search radius and a nucleus density analysis map of traditional villages in the Awa Mountain area was derived (Figure 2). The analysis concluded that the traditional villages in the northwest area are densely distributed, and two polar nuclei appear in space as a whole, with a scattered distribution in the northwest and southwest. Fengqing (10), and Jiangcheng (8) are the two areas with extremely high-density polar nucleus regions. The first of the two "sides" was Linxiang (7), Canyuan (6), Jingdong (5), Yongde (5), Zhenyuan (4); the second "side" was Lancang (7) and Menglian (5). The other scattered sites were randomly distributed.

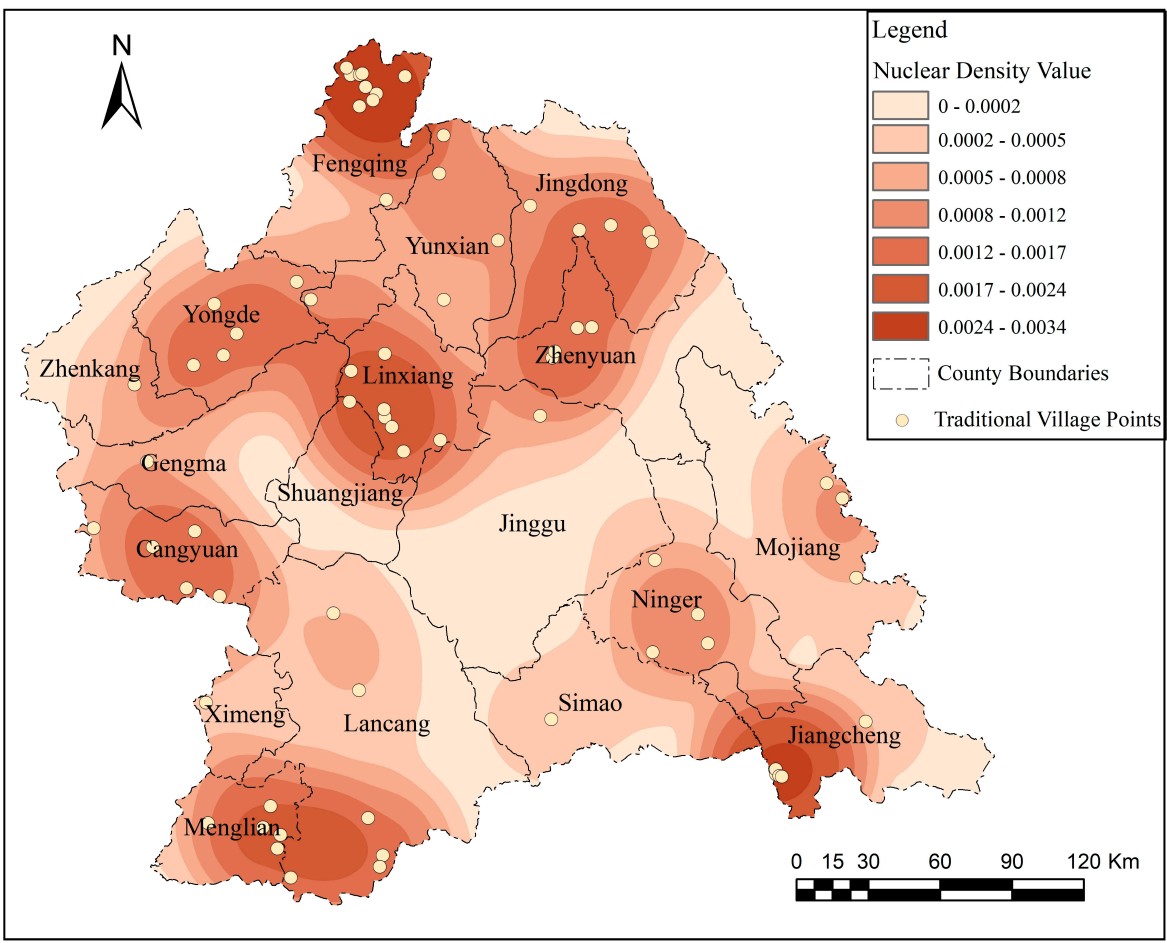

**Figure 2.** Nuclear-density density-map of traditional villages in Awa Mountain.

*4.2. Spatial Autocorrelation Analysis*

Spatial autocorrelation is used to check whether there is a correlation between the study elements and their neighboring elements, and spatial autocorrelation is divided into global autocorrelation and local autocorrelation. In order to understand the spatial distribution of traditional villages in the Awa Mountain area more clearly, this paper uses global Moran's I and local Moran's I for analysis [37].

4.2.1. Global Moran's I

The spatial weight matrix was created by the K-nearest neighbor method [38] in the Geoda software with a Moran's I of 0.774 (Figure 3), and the data were discretized after 999 permutations with a *p*-value of 0.001 (less than 0.01 was significant) and a Z-value of 11.0692, indicating that there is a positive correlation in the overall spatial distribution of traditional villages in the Awa Mountain area.

4.2.2. Partial Moran's I

Since the global spatial autocorrelation cannot reflect the heterogeneity of the local area, further analysis was conducted with local Moran's I to generate a map of traditional villages in the Awa Mountain area (Figure 4). Within the counties in the study, Linxiang belongs to a high–high (H–H) agglomeration, indicating its high traditional village density and strong spatial connectivity; Yunxian, Jingdong, and Canyuan belong to a high–low (H–L) agglomeration, which is a region with high traditional village density but not strong spatial connectivity; Zhenkang, Gengma, and Ning'er belong to low–high (L–H) clusters, which are traditional villages with low density but strong spatial connectivity; and only 11%

of traditional villages showed low–low clusters. This indicated that there are differences in the spatial distribution of local areas, which were further analyzed by the GWR model.

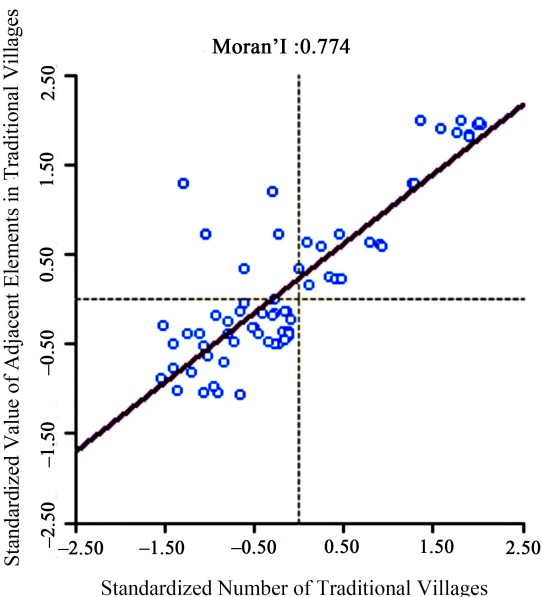

**Figure 3.** Global Moran's I diagram of traditional villages in Awa Mountain.

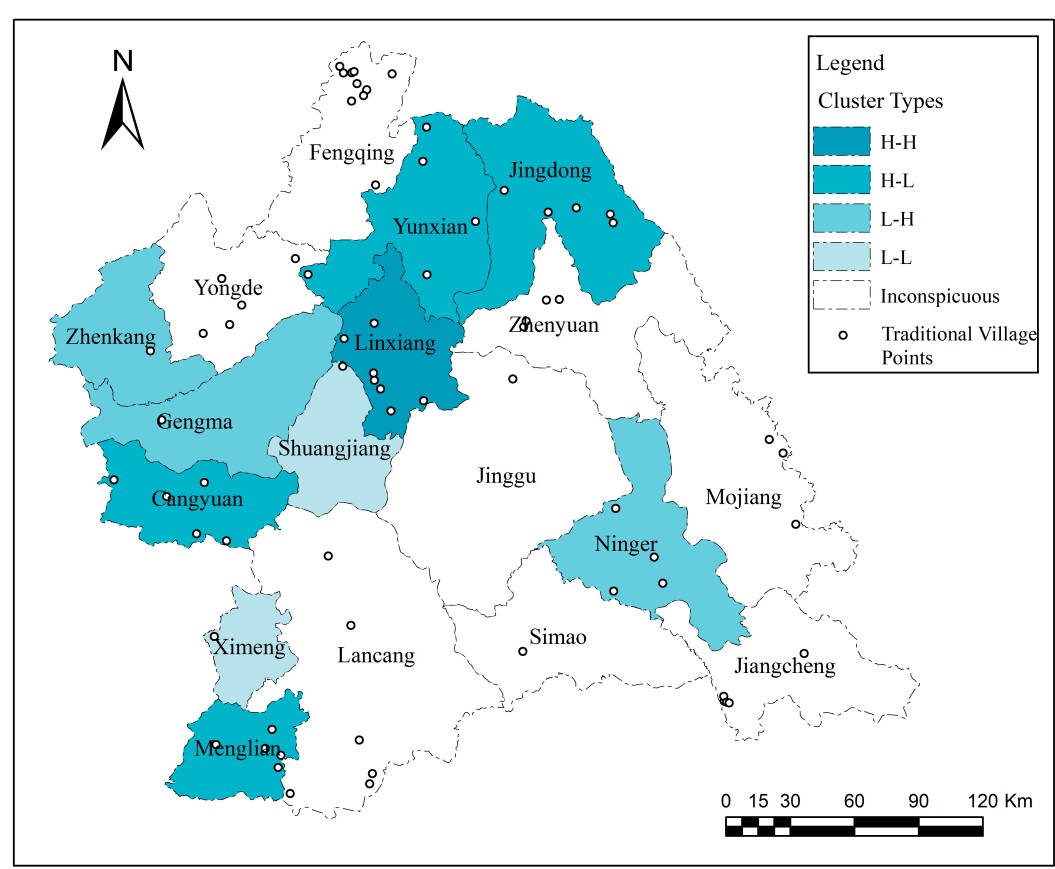

**Figure 4.** Partial Moran's I diagram of traditional villages in Awa Mountain.

*4.3. Spatial Heterogeneity Analysis*

4.3.1. OLS Model

In this study, elevation (X1), slope (X2), slope aspect (X3), average annual precipitation (X4), average annual temperature (X5), distance from a road (X6), distance from a river

(X7), population density (X8), GDP (X9), percentage of ethnic minorities (X10), number of intangible cultural heritage units (X11), and the number of protected cultural relics (X12) were chosen as the independent variables, and the nuclear density of traditional villages Nuclear (Y) was used as the dependent variable to construct the regression model. The data of the 12 factors were standardized and imported into ArcGIS10.2 for OLS model analysis (Table 1) [39]. The $R^2$ and corrected $R^2$ of the OLS model were 0.52 and 0.42, respectively, and the $AIC_C$ value was 853.52. Six variables, including elevation (X1), average annual precipitation (X4), population density (X8), percentage of ethnic minorities (X10), number of intangible cultural heritage units (X11), and the number of protected cultural heritage units (X12) passed the significance and covariance tests. The results showed that natural factors, spatial factors, and regional cultural factors are significantly correlated with the spatial distribution of traditional villages. There are still some factors with insignificant correlations, which were not consistent with our expectations, indicating the complexity of the influential effects of the factors, which need to be further analyzed and illustrated with the actual situation of the study area.

**Table 1.** OLS model result analysis table.

| Independent Variables | Coefficient | Standard Deviation | VIF | Robust_SE | Robust_t | Robust_Pr |
|---|---|---|---|---|---|---|
| X1 | −273.341 | 85.626 | 12.592 | 92.701 | −2.949 | 0.004 * |
| X2 | 17.402 | 14.789 | 1.560 | 17.015 | 1.023 | 0.310 |
| X3 | 2.932 | 7.425 | 1.202 | 6.406 | 0.458 | 0.649 |
| X4 | −237.565 | 67.673 | 1.429 | 54.400 | −4.367 | 0.000 * |
| X5 | −328.153 | 253.563 | 12.340 | 277.860 | −1.181 | 0.242 |
| X6 | −3.298 | 3.059 | 1.209 | 3.210 | −1.027 | 0.308 |
| X7 | −4.482 | 7.415 | 1.513 | 6.651 | −0.674 | 0.503 |
| X8 | −20.081 | 8.962 | 2.260 | 9.268 | −2.167 | 0.034 |
| X9 | 13.005 | 47.187 | 1.837 | 39.930 | 0.326 | 0.746 |
| X10 | −101.874 | 23.956 | 3.140 | 20.990 | −4.854 | 0.000 * |
| X11 | −0.310 | 3.186 | 2.030 | 2.521 | −0.123 | 0.902 |
| X12 | 28.125 | 5.460 | 1.580 | 4.642 | 6.060 | 0.000 * |

"*"represents the statistically significant *p* value ($p < 0.05$).

### 4.3.2. GWR Modeling

The GWR modeling clearly showed the heterogeneity of the spatial distribution of each element in the traditional villages in the Awa Mountain area (Table 2). Geographically weighted regression analysis was conducted in ArcGIS10.2 with the AICc value as the optimal bandwidth. The results showed that the mean values of the regression coefficients for each factor had positive and negative values, indicating great spatial variability, and the larger regression coefficient values indicated a greater influence and a stronger effect on the spatial distribution(Figure 5). The $R^2$ and corrected $R^2$ of the GWR modeling were 0.81 and 0.73, respectively, and the $AIC_C$ value decreased to 158.485, indicating the better fit of the GWR model, which was used for further analysis.

**Table 2.** GWR model results analysis table.

| Independent Variables | Est. | Robust_SE | Robust_t | *p*-Value | Standard Deviation |
|---|---|---|---|---|---|
| X1 | −0.999 | 0.313 | −3.192 | 0.001 * | 0.755 |
| X2 | 0.130 | 0.110 | 1.177 | 0.239 | 0.028 |
| X3 | 0.038 | 0.097 | 0.395 | 0.693 | 0.179 |
| X4 | −0.370 | 0.105 | −3.510 | 0.000 * | 0.071 |
| X5 | −0.409 | 0.316 | −1.294 | 0.196 | 0.494 |
| X6 | −0.105 | 0.097 | −1.078 | 0.281 | 0.106 |
| X7 | −0.066 | 0.108 | −0.604 | 0.546 | 0.088 |
| X8 | −0.297 | 0.133 | −2.241 | 0.025 | 0.088 |
| X9 | 0.033 | 0.120 | 0.276 | 0.783 | 0.136 |
| X10 | −0.664 | 0.156 | −4.253 | 0.000 * | 0.084 |
| X11 | −0.012 | 0.125 | −0.097 | 0.922 | 0.125 |
| X12 | 0.571 | 0.111 | 5.152 | 0.000 * | 0.196 |

"*"represents the statistically significant *p* value ($p < 0.05$).

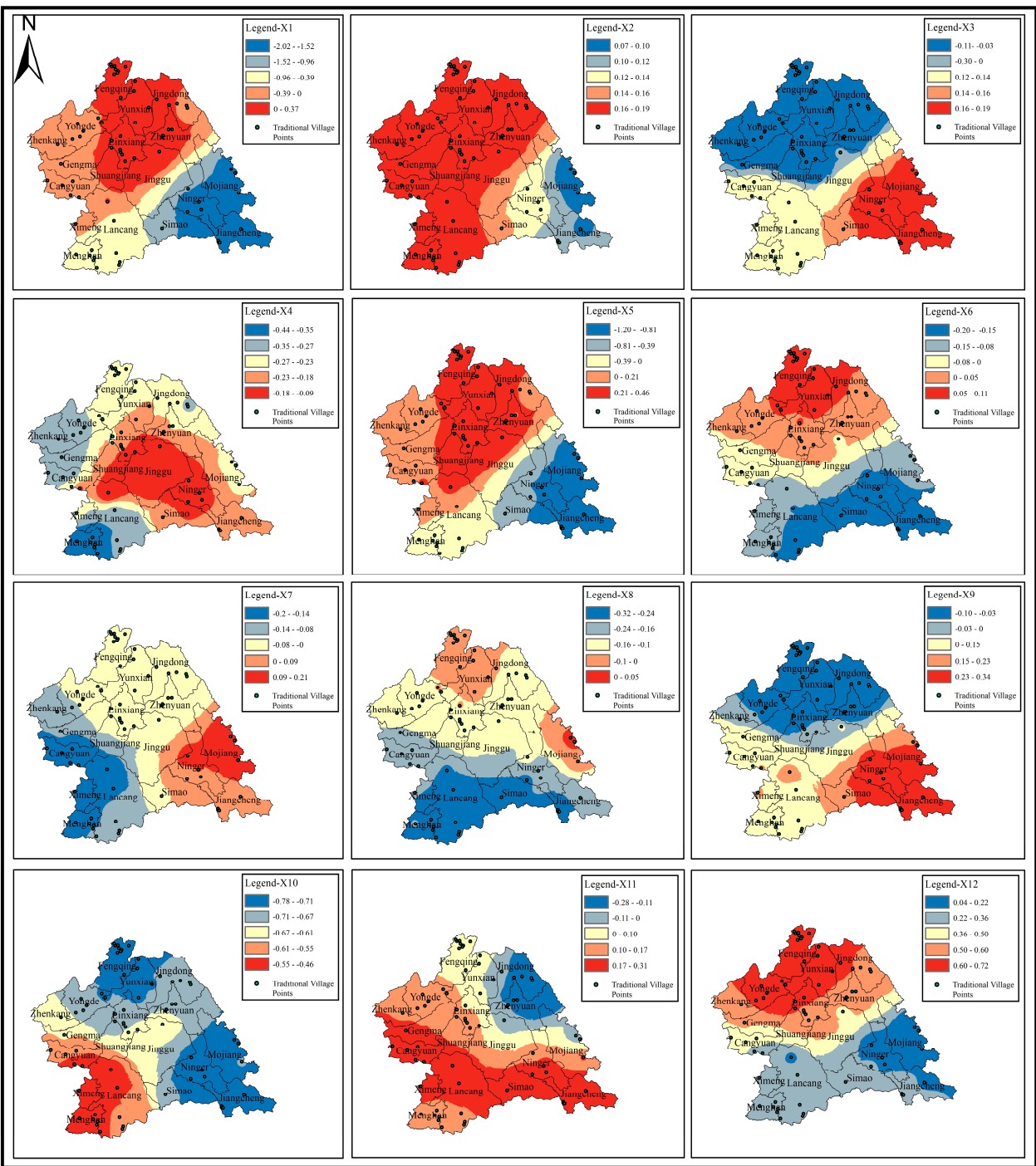

**Figure 5.** GWR model coefficient analysis diagram of traditional villages in Awa Mountain.

(1) Natural factors

Elevation (X1) had both a positive and negative influence on the spatial distribution of traditional villages in the Awa Mountain area, with the overall distribution pattern of being high in the northwest and low in the southeast, and the regression coefficients ranged from −2.02 to 0.37, with a large difference in the average coefficient values, indicating strong spatial variability. According to the altitude classification standard in China, 72% of the traditional villages in the study area were distributed at a medium altitude from 1000 m to 2000 m. Elevation was positively associated with the distribution of traditional villages in the northwest, indicating that higher elevation in the northwest had a greater influence on the distribution of traditional villages. This also proves that the traditional villages are

less damaged and relatively intact in terms of conservation due to the formation of closed environmental conditions in relatively high-altitude areas.

The slope (X2) had a positive influence on the spatial distribution of traditional villages in the Awa Mountain area, and the influence decreased from west to east. According to the Technical Regulations of the Third National Land Survey (TD/T1055-2019) and the standard paradigm, the slope of arable land is classified as Grade I (<2), Grade II (2–6), Grade III (6–15), Grade IV (15–25), and Grade V (>25), and the number of traditional villages in each category in the Awa Mountain area is 2.7, 20, 58.7, 16, and 2.76, respectively. It can be seen that traditional villages are mainly distributed on slopes of 6–15° slope. Because the study area is mountainous, the village sites had a limited range of slopes to choose from, and more residents chose areas with a relatively gentle slope according to the soil conditions of cultivated land. In addition, steeper slopes can also avoid flooding problems to a certain extent [40].

The influence of the slope aspect (X3) on the spatial distribution of traditional villages in the Awa Mountain area decreased from southeast to northwest, and the regression coefficient ranged from $-0.11$ to 0.41, with 35.1% ($p \leq 0.05$) of the traditional villages having a significant correlation with slope aspect. In the Northern Hemisphere, slopes with 90–270° are considered to be sunny with good light conditions, and the rest are shady slopes. In total, 53.3% of the villages are distributed on sunny slopes, and the role of slope orientation in the mountain habitat is more important than in plains. The location of traditional villages on shady slopes is related to their complex topography or proximity to water sources, which proves that the spatial distribution of traditional villages is complicated by multiple factors.

The regression coefficient of average annual precipitation (X4) decreased from the central part to the surrounding area, indicating that the spatial distribution of traditional villages in the central area was more influenced by precipitation, while the areas at the edge were less influenced by precipitation, and 70.1% ($p \leq 0.05$) of traditional villages in the Awa Mountain area were significantly correlated with the annual precipitation factor. The precipitation trend decreased gradually from west to east, and the annual precipitation was between 856–1151 mm. The study area belongs to the humid zone, and precipitation is one of the necessary factors for carrying out farming activities. The eastern area, with less precipitation, has fewer farming-type settlements, while the central and western areas promoted agricultural development because of the suitable climatic conditions.

The annual mean temperature (X5) had both a positive and negative influence on the spatial distribution of traditional villages in the Awa Mountain area, and the regression coefficient ranged from $-1.2$ to 0.46, with a positive influence in the northwest and a negative influence in the southeast, with obvious spatial heterogeneity. The average annual temperature ranged from 14.2 °C to 23.3 °C, the number of traditional villages distributed in the temperate zone of 17–18.5 °C was the largest, the suitable temperature is suitable for human habitation and crop growth, it also promotes the formation of traditional villages, and fewer traditional villages distributed in the southwestern region where the temperature was lower.

(2) Spatial factors

The regression coefficient of distance from the road (X6) ranged from $-0.2$ to 0.11, which has a positive influence on the traditional villages in the northern part of the Awa Mountain area, decreasing from north to south. Fengqing and Yunxian had a relatively large number of traditional villages because of their higher altitude, complex topography and lower road network density, and they were not easily disturbed and assimilated by the outside world, so the traditional villages have been preserved more completely, though this has also constrained the economic development of the area. The development of the regional economy has been hampered. However, the number of traditional villages in southern areas such as Simao and Ning'er, where the road network density is relatively high, is relatively large, which is conducive to population mobility and increased external

communication. These areas face greater opportunities and challenges for the development of traditional villages.

The regression coefficient value of the distance from a river (X7) decreased from the east to the west. The water system in the eastern region of the Awa Mountain area is more developed than that in the western region, providing a favorable guarantee for the development of traditional villages in the east. The location and layout of traditional villages have always been based on mountains and water, and water has a greater influence on the villagers' productivity and life. Unlike the eastern traditional villages, the western Canyuan, Ximeng, and Menglian are prone to flooding because of the high annual precipitation and the low density of the river network, so for safety, most of the traditional villages in the western areas are distributed at a certain distance from the main rivers and large water bodies.

(3) Social factors

Population density (X8) had a positive influence on the spatial distribution of traditional villages in the northern part of the Awa Mountain area, which was related to the relatively high density of traditional villages in this region. There are fewer traditional villages in the southern Lancang and Simao because of the larger land area and the relatively less concentrated population. A low regression coefficient was found in Jiangcheng, which have a low population density but a large number of traditional villages. This also indicates that a lower population density will reduce the damage to the landscape of traditional villages caused by large-scale population migration, and also indicates that the problem of "hollowing out" traditional villages is still serious. How to revitalize the villages and improve their attractiveness is a problem to be solved in the process of urbanization.

The regression coefficient of GDP (X9) decreased from the southeast to the north. The traditional villages are more densely distributed in the regions with lower economic development, and the areas with lower economic development are less affected by population movements and urbanization, so that the traditional village culture is preserved more completely. However, the high density of traditional villages in Jiangcheng in the area with a highly positive impact proves that the development of traditional villages and the economy in this area has formed a good promotion relationship. The population of ethnic minorities in this county accounts for 79.8% of the total population, which proves that the factors influencing the spatial distribution of traditions are complex.

(4) Regional cultural factors

The regression coefficient of the percentage of ethnic minorities (X10) decreased from southwest to northeast, and 95.1% ($p \leq 0.05$) of the traditional villages were significantly correlated with this factor. The areas where this factor had a strong influence were found in Canyuan, Ximeng, and Menglian in the southwest, and the percentage of their ethnic minority population is 94, 91, and 86, respectively, mainly the two cross-border Wa populations. This also indicates that the spatial distribution of ethnic minority areas and traditional villages in the Awa Mountain area are closely related. The ethnic minorities in this area are mainly related to ethnic groups that have been living together for a long time due to clan relationships, and that have their own independent production and lifestyle so that their uniqueness has been preserved. To a certain extent, the development of traditional villages has been promoted.

The regression coefficient of the intangible cultural heritage (X11) showed a positive effect, with 82.7% ($p \leq 0.05$) of traditional villages significantly correlated with this factor, decreasing from southwest to northeast overall, with the areas with a highly positive effect appearing in Lancang, Canyuan, Simao, Jiangcheng Hani Yi Autonomous County, etc. These areas are geographically close to other countries and have a strong minority culture, which brings more folklore to the traditional villages, and many of them rely on intangible cultural heritage to drive economic development.

The regression coefficient of the number of protected cultural relic units (X12) decreased from south to north, and there was an overall positive effect, as 86.7% ($p \leq 0.05$) of traditional villages in the Awa Mountain area had a significant correlation with this factor,

with a denser distribution of traditional villages in the northern region. The main types of protected cultural relic units in the region are ancient buildings and ancient sites, which are related to factors of the historical economy and development of trade factors.

### 4.3.3. Geodetector Results

(1) Dominant factors

The q-values of the explanatory power of each factor on the spatial distribution of traditional villages in the Awa Mountain area are, in descending order, the proportion of ethnic minorities (X10) > the intangible cultural heritage (X11) > GDP (X9) > average annual temperature (X5) > elevation (X1) > average annual precipitation (X4) > the number of protected cultural relic units (X12) > slope (X2) > total population (X8) > distance from rivers (X7) > slope aspect (X3) > distance from roads (X6) (Figure 6, Table 3). This indicates that the proportion of minority populations (X10) is the main driving factor affecting the spatial distribution of traditional villages in the Awa Mountain area, with an explanatory power of 0.54. Among the social factors, GDP (X9) had the strongest explanatory power of 0.42; elevation (X1) had the strongest explanatory power among all the natural factors, with an explanatory power of 0.31. Among the natural factors, the slope aspect and distance from a road had relatively weak explanatory power, and the results were closely related to the geographic environment and history of the Awa Mountain area. Here, the topography is suitable for agricultural production, and the water supply is adequate, which were the primary conditions of site selection to ensure the villagers' livelihood, as opposed to being on a sunny slope. In terms of spatial location, because traditional villages are generally minority areas, they have less communication with the outside world and do not like to be disturbed by the outside world, and these complex geological conditions have led to the road network density in southwest Yunnan being relatively low.

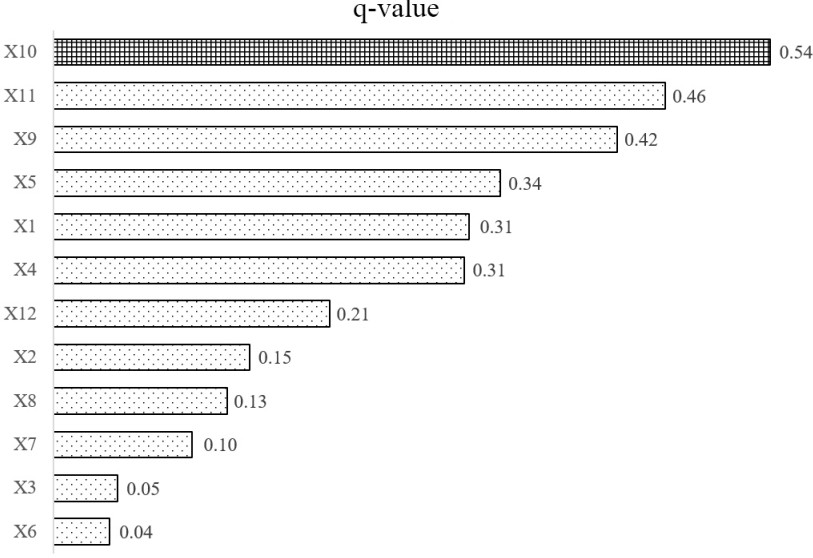

**Figure 6.** q-value ranking chart.

(2) Interactive factors

In this study, the GWR modeling was used to analyze the individual influencing factors of traditional villages' distribution in the Awa Mountain area. Still, the results showed that the distribution of traditional villages was influenced by a complex interaction of factors. The formation of the villages was influenced by a combination of natural, spatial, economic and regional cultural factors driving their distribution (Table 4) (Figure 7). Further analysis was performed with the interaction detector to detect the interactions among the factors. The overall nonlinear growth types were highest with two factors, in which the interaction of annual precipitation (X4) ∩ minority share (X10) was the strongest interactive driver

of nonlinear growth, with an explanatory power of 0.93. This also proved that the most significant driving combinations were found for two independent variables, and annual precipitation (X4) ∩ minority share (X10), annual precipitation (X4) ∩ intangible cultural heritage (X11), and annual precipitation (X4) ∩ the number of protected cultural relic units (X12) were the three subsets with the strongest explanatory driving force. However, the explanatory power of X3 ∩ X9 tended to weaken, which is consistent with the previous results of dominant factor detection.

**Table 3.** Statistical table of dominant driving forces and significance.

| Independent Variables | q-Value | Sig |
|---|---|---|
| x1 | 0.313 | 0.007 |
| x2 | 0.148 | 0.017 |
| x3 | 0.048 | 0.765 |
| x4 | 0.309 | 0.002 |
| x5 | 0.337 | 0.001 |
| x6 | 0.043 | 0.736 |
| x7 | 0.104 | 0.308 |
| x8 | 0.131 | 0.201 |
| x9 | 0.425 | 0.000 |
| x10 | 0.540 | 0.000 |
| x11 | 0.461 | 0.000 |
| x12 | 0.208 | 0.038 |

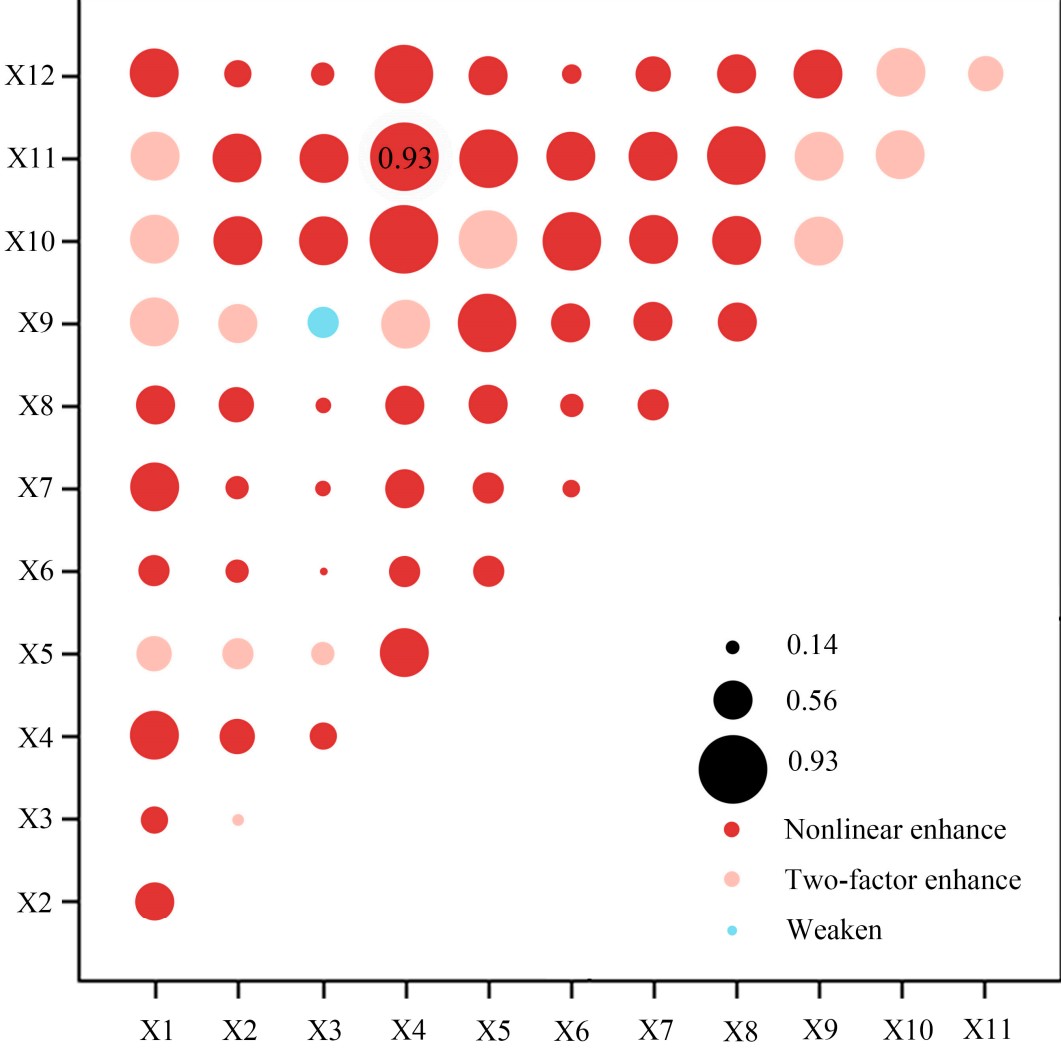

**Figure 7.** Interactive factor detection diagram of traditional villages in Awa Mountain.

**Table 4.** Interactive factor detection table.

| Independent Variables | X1 | X2 | X3 | X4 | X5 | X6 | X7 | X8 | X9 | X10 | X11 | X12 |
|---|---|---|---|---|---|---|---|---|---|---|---|---|
| X1 | - | - | - | - | - | - | - | - | - | - | - | - |
| X2 | 0.538 | - | - | - | - | - | - | - | - | - | - | - |
| X3 | 0.365 | 0.155 | - | - | - | - | - | - | - | - | - | - |
| X4 | 0.699 | 0.463 | 0.155 | - | - | - | - | - | - | - | - | - |
| X5 | 0.555 | 0.472 | 0.345 | 0.155 | - | - | - | - | - | - | - | - |
| X6 | 0.455 | 0.352 | 0.137 | 0.460 | 0.155 | - | - | - | - | - | - | - |
| X7 | 0.675 | 0.352 | 0.259 | 0.578 | 0.460 | 0.155 | - | - | - | - | - | - |
| X8 | 0.546 | 0.511 | 0.218 | 0.587 | 0.580 | 0.314 | 0.155 | - | - | - | - | - |
| X9 | 0.730 | 0.547 | 0.418 | 0.701 | 0.827 | 0.538 | 0.568 | 0.155 | - | - | - | - |
| X10 | 0.677 | 0.711 | 0.624 | 0.930 | 0.839 | 0.782 | 0.718 | 0.723 | 0.155 | - | - | - |
| X11 | 0.752 | 0.690 | 0.610 | 0.929 | 0.804 | 0.661 | 0.669 | 0.738 | 0.698 | 0.155 | - | - |
| X12 | 0.618 | 0.376 | 0.321 | 0.838 | 0.552 | 0.277 | 0.466 | 0.584 | 0.686 | 0.653 | 0.155 | - |

## 5. Discussion

Through the joint efforts of the state and all of society, traditional villages have been effectively protected, but problems have still emerged during their development, such as the serious problem of hollowing out villages through excessive development of tourism [41], the loss of architectural elements of dwellings resulting from low ethnic cultural identity [42], the impact of multiculturalism on ethnic cultural identity [43], and the loss of primitive worship of natural landscapes by many ethnic minorities resulting from modern farming techniques [44]. In previous studies, Yang Yan et al. [45] (2021) considered the natural environment as a fundamental factor for studying the spatial distribution of traditional Miao villages in Guizhou Province, with population and culture playing a decisive role, and economy and transportation playing a protective role. Nie Xiaokang et al. [46] (2022) studied the Gansu section of the Bailong River basin to reveal the layout of traditional villages, revealing the strong influence of rivers and slopes. Yin Wei et al. [47] (2022) studied the spatial distribution characteristics of traditional villages in the Bashu Cultural Corridor (Chengdu–Chongqing area) and found that natural factors influenced the origin, human factors influenced the development, and various factors jointly constrained the development. In summary, different research backgrounds have led to different results. The most essential difference is that the research areas are different in terms of geography and landscape, leading to variability in the village sites, the selection of the sites that promoted or restricted the development of villages, and the differences in culture and economy that led to different major industries.

The spatial distribution of traditional villages in the Awa Mountain area is essentially influenced by natural factors, spatial factors, social factors, and regional cultural factors, and the mechanism is dynamic and complex with multi-factor effects. The spatial location preference of traditional villages in Awa mountain area can be obtained by reclassifying 12 independent variables from four categories of factors (Table 5). Elevation (X1) and average annual precipitation (X4) are significantly correlated. The dominant factors had an explanatory power of 0.31. The degree of interaction detected for average annual precipitation (X4), ∩ average annual temperature (X5), and elevation (X1), ∩ average annual precipitation (X4) had relatively higher explanatory power, reaching 0.73 and 0.7. This proved that the explanatory power of multiple factors was much greater than that of single factors, in which average annual precipitation (X4) ∩ number of intangible cultural heritage (X11) was the subset with the strongest global explanatory power, proving that natural factors and regional cultural factors were the main driving forces. Natural factors are the basic factors determining the distribution of traditional villages in the Awa Mountain area [48]. The Awa Mountain area has mainly farming villages, with a suitable climate, fertile soil, a dense river network, and sufficient light benefiting the living environment of traditional villages. The elevation, slope, slope aspect, average annual precipitation, and average annual temperature have determined the basic spatial distribution patterns

of traditional villages but also, to some extent, have restricted the scale and pattern of expansions. The effects of distance from a road (X6) and distance from a river (X7) were not very obvious, and distance from a river (X7) ∩ elevation (X1) had a q-value of 0.68, which has the strongest explanatory power from the perspective of spatial factors, proving that spatial factors and natural factors jointly drive changes in the spatial patterns of traditional villages. Distance from water sources (X6), slope (X2), and slope aspect (X3) contributed to and constrained the basic site layout of the villages. The spatial factors played a supplementary role because the high altitude and complex terrain of the Awa Mountain area restricted transportation. There are many kinds of ethnic minorities and different ethnic habits, so coupled with the strong control of ethnic minority leaders and exploitation by Han landlords, there was less communication with the outside world, which inhibited the economic development of the villages and less impact from modernization so that the excellent historical and cultural landscape has been preserved [49]. Population density (X8) had a significant correlation, and GDP (X9) was the main driver, with an explanatory power of 0.42, whereas population density (X8) ∩ intangible cultural heritage (X11) was the strongest subset of socioeconomic factors in terms of explanatory power. Economic factors have played a decisive role in two ways [50]. On the one hand, the economy as a whole showed a negative correlation, with the backward economy playing an important role in the protection of traditional villages. On the other hand, Jiangcheng Hani-Yi Autonomous County showed a positive correlation, proving that the region has formed a good symbiotic relationship. In terms of population, a low population density reduces the impact of modernization on traditional villages but inhibits economic development. The proportion of ethnic minorities (X10), the intangible cultural heritage (X11), and the number of protected cultural heritage units (X12) were significantly correlated, with X10 and X11 being the two factors with the strongest explanatory power. The regional cultural factors are key factors, the regional cultural factors have played a strong role in promoting and dominating the spatial distribution of traditional villages, and the factors complement each other [51]. Traditional villages nurture diverse minority cultures, and, together with the geographical proximity to other countries, the intangible cultural heritage drives the local economic development.

**Table 5.** Reclassification table of traditional villages in the Awa Mountain area.

| Index Factor | Number | Standard of Classification | Classification Result | | |
|---|---|---|---|---|---|
| | | | Name | Quantity (units) | Proportion (%) |
| altitude | X1 | <1 km is low altitude, 1 km–2 km is medium altitude, 2 km–4 km is above average altitude, 4 km–6 km is high altitude, and >6 km is extremely high altitude; | low altitude | 14 | 18.6 |
| | | | medium altitude | 54 | 72 |
| | | | above average altitude | 7 | 9.4 |
| slope | X2 | <2 is Grade I, 2–6 is Grade II, 6–15 is Grade III, 15–25 is Grade IV, and >25 is Grade V; | Grade I | 2 | 2.6 |
| | | | Grade II | 15 | 20 |
| | | | Grade III | 44 | 58.7 |
| | | | Grade IV | 12 | 16 |
| | | | Grade V | 2 | 2.7 |
| aspect | X3 | 90–270 degrees is a sunny slope, and the rest is a shady slope; | sunny slope | 40 | 53.3 |
| | | | shady slope | 35 | 46.7 |
| annual precipitation | X4 | Natural | 856–997 mm | 20 | 26.7 |
| | | | 997–1104 mm | 18 | 24 |
| | | | 1104–1215 mm | 21 | 28 |
| | | | 1215–1330 mm | 7 | 9.3 |
| | | | 1330–1551 mm | 9 | 12 |
| annual mean temperature | X5 | Natural | 14.2–15 °C | 2 | 2.7 |
| | | | 15–17 °C | 14 | 18.7 |
| | | | 17–18.5 °C | 24 | 32 |
| | | | 18.5–20 °C | 18 | 24 |
| | | | 20–23.3 °C | 17 | 22.6 |

**Table 5.** *Cont.*

| Index Factor | Number | Standard of Classification | Classification Result | | |
|---|---|---|---|---|---|
| | | | Name | Quantity (units) | Proportion (%) |
| distance from road | X6 | Natural | 0–1 km | 62 | 82.7 |
| | | | 1–2 km | 4 | 5.4 |
| | | | 2–6 km | 6 | 8 |
| | | | 6–8 km | 1 | 1.3 |
| | | | >8 km | 2 | 2.6 |
| distance from river | X7 | Natural | 0–1 km | 19 | 25.3 |
| | | | 1–2 km | 10 | 13.3 |
| | | | 2–3 km | 15 | 20 |
| | | | 3–6 km | 19 | 25.3 |
| | | | >6 km | 12 | 16 |
| population density | X8 | Natural | 0–100 person/km$^2$ | 42 | 56 |
| | | | 100–200 person/km$^2$ | 10 | 13.3 |
| | | | 200–300 person/km$^2$ | 10 | 13.3 |
| | | | 300–1000 person/km$^2$ | 9 | 12 |
| | | | >1000 person/km$^2$ | 4 | 5.3 |
| GDP | X9 | Natural | 22,963–25,000 yuan/person | 5 | 6.7 |
| | | | 25,000–30,000 yuan/person | 18 | 24 |
| | | | 30,000–38,000 yuan/person | 30 | 40 |
| | | | 38,000–45,000 yuan/person | 21 | 28 |
| | | | 45,000–54,726 yuan/person | 1 | 1.3 |
| proportion of ethnic minorities | X10 | Natural | >30% | 62 | 82.7 |
| | | | ≤30% | 13 | 17.3 |
| number of intangible cultural heritage | X11 | Natural | >3 | 42 | 56 |
| | | | ≤3 | 33 | 44 |
| number of cultural relics protection units | X12 | Natural | >3 | 20 | 26.7 |
| | | | ≤3 | 55 | 73.3 |

## 6. Conclusions

In this study, 75 traditional villages in the Awa Mountain area were taken as the research object, the spatial distribution characteristics and driving factors of traditional villages in Awa Mountain area were analyzed by using GWR model and geodetector, which provided quantitative research support for the development of traditional villages. The main conclusions were as follows:

(1) As a whole, there were two polar nuclei in the space of the Awa Mountain area, northwest, and southwest, which were scattered. Taking Fengqing and Jiangcheng as two high-density extreme areas, the global space presented a significant positive correlation, Moran, I is 0.774, and there were two types of spatial agglomeration in the central area of Awa Mountain, as well as 55.6% of the traditional villages were clustered.

(2) In the Awa mountain area, 72% of the traditional villages were located in the middle altitude area of 1000–2000 m, 44% of the traditional villages were located on the sunny slope of 90–270, and the annual precipitation zone of 78.7% of the traditional villages was in the range of 856–1215 mm. The results also showed that 82.7% of the traditional villages are within 1 km of the road, 25.3% of the traditional villages were within 1 km of the water source, 56% of the traditional villages have a population density of 0–100 people/km, and 40% of the traditional villages have a GDP of 30,000–38,000 yuan/person.

(3) Natural factors were the basic factors that affect the spatial distribution of traditional villages in the Awashan area, and determine the location and development scale of villages; spatial factors were auxiliary factors; social factors were decisive factors, which were negatively correlated globally and positively correlated locally; regional culture was the key factor, and the two complement each other; factors such as

backward economic level, blocked traffic, less external communication, and low population density has been playing a protective role.

(4) The results of the Geodetector revealed that the proportion of minority population (X10) was the main driving factor that affects the spatial distribution of traditional villages in the Awashan area, with q-values reaching 0.54; among social factors, GDP(X9) has the strongest explanatory power, reaching 0.42; elevation (X1) was the strongest explanatory power among all natural factors, with explanatory power of 0.31, annual precipitation (X4) ∩ minority proportion (X10), and the nonlinear interaction driving force was the strongest, with q-values reaching 0.93. At the same time, it was proven that when the number of independent variables was 2, it was the most significant driving combination.

This study indicates the following suggestions based on the research results. Firstly, cultural revitalization is the only way to revitalize villages, as cultural identity and cultural confidence enhance the attractiveness of villages. Regional cultural factors play a significant role in the spatial distribution of traditional villages in the Awa Mountain area, so protecting the regional culture will protect the development of ethnic minorities. Multiple measures can be taken to realize cultural revitalization, encouraging more young talents to participate in the development and construction of traditional villages, tapping the enthusiasm for building villages. Second, we should strengthen the construction of infrastructure to promote high-quality economic development and industrial integration. Social factors have a decisive influence, as the economy drives the development of villages, tourism stimulates the development of villages. The intangible cultural heritage is an important and valuable resource for the development of traditional villages, and thus, reasonable tourism development to promote the national culture can also drive economic development, create good conditions, fully explore the combination of regional culture and tourism, and promote industrial integration and upgrading. Third, we can improve villagers' living environment. Natural factors are basic factors. Natural disasters such as mudslides, floods and fires often occur in the Awa Mountain area, so we should strengthen the protection of the village landscape, especially ensuring residential security, further improving the public infrastructure of traditional villages. Fourth, we should improve the transportation network, as transportation plays an important role. The complex terrain of the Awa Mountain area and the restricted transportation have hindered the development of villages, so we should actively promote the interconnection of road networks to increase communication with the outside world. Transportation also drives economic and cultural development, so villages with high road network density should maintain their ethnic characteristics and alleviate the impact of modernization. Lastly, protection strategies should be developed according to the classification of the villages' influencing mechanisms. The spatial distribution of traditional villages is the result of the coupled effect of multiple factors, and the spatial variability of different factors is obvious and different. For example, in traditional village clusters, such as in Fengqing in the northern part of the study area, on the basis of not destroying the overall appearance and cultural authenticity of the villages, the development of traditional villages and local industrial development can be organically integrated to achieve a better symbiotic relationship between people and villages, such as the appropriate development of tourism and cultural products, handicrafts, folklore and cultural performances, cultural and creative products, and characteristic folklore.

The factors affecting the distribution of traditional villages are very complex and may require data from other disciplines to jointly support the research. Not enough information on the history of the Awa Mountain area has been collected, and it is necessary to continue to study the patterns of the landscape, residential characteristics, street spaces, and cultural archaeology of traditional villages from a microscopic perspective in the future.

**Author Contributions:** S.L. performed the data analysis and wrote the manuscript, Y.S., H.X., Y.L. and S.Z. corrected our manuscripts. All authors have read and agreed to the published version of the manuscript.

**Funding:** National Natural Science Foundation of China (51968064), "Yunnan Province High-level Talents Training Support Program" (YNWR-CYJS-2020-022), Yunnan University Minority Gardens, and beautiful countryside Science and Technology Innovation Team.

**Institutional Review Board Statement:** Not applicable.

**Informed Consent Statement:** Not applicable.

**Data Availability Statement:** Not applicable.

**Conflicts of Interest:** The authors declare no conflict of interst.

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
