# Peer review of "Spatial Distribution Characteristics and Driving Factors for Traditional Villages in Areas of China Based on GWR Modeling and Geodetector: A Case Study of the Awa Mountain Area"

_sustainability, doi:10.3390/su15043443_

Round 1

Reviewer 1 Report (Previous Reviewer 1)

Dear author/authors,

It is an interesting study based on a complex methodology.

Despite the complex methodology that combines natural, social, spatial, cultural factors, the technical part is too much detailed. This fact can cause difficulties for the readers in order to understand differences identified at the level of traditional villages. It would be necessary to create a typology of traditional villages, since the study relies on an impressive amount of quantitative data.

In this sense, the discussion part should capture these differences/similarities. The data from Table 1 could be very helpful in this interpretation and, in the end, the table could/should be eliminated.

Conclusions have also to be rewritten. The number of references is insignificant. Please add other publications.

Pay attention to the bibliography style and respect the requirements of the journal.

Best regards.

Author Response

Reviewer 2 Report (Previous Reviewer 2)

In figure 4, on page 8, the legend regarding Linxiang appears erroneously as L-L and should be, according to calculations and text, as H-H.

Author Response

Reviewer 3 Report (Previous Reviewer 4)

Accept as it is.

Round 2

Reviewer 1 Report (Previous Reviewer 1)

I agree with the corrections. 

This manuscript is a resubmission of an earlier submission. The following is a list of the peer review reports and author responses from that submission.

Round 1

Reviewer 1 Report

Dear author/authors,

It is an interesting study based on a complex methodology.

Despite the complex methodology that combines natural, social, spatial, cultural factors, the technical part is too much detailed. This fact can cause difficulties for the readers in order to understand differences identified at the level of traditional villages. It would be necessary to create a typology of traditional villages, since the study relies on an impressive amount of quantitative data.

In this sense, the discussion part should capture these differences/similarities. The data from Table 1 could be very helpful in this interpretation and, in the end, the table could/should be eliminated.

Conclusions have also to be rewritten. The number of references is insignificant. Please add other publications.

Pay attention to the bibliography style and respect the requirements of the journal.

Best regards.

Reviewer 2 Report

The text is written concisely and clearly. The number of variables under analysis is large, which makes reading the text somewhat time-consuming.

The description of the mathematical tools used is complex and requires careful reading, but it is always coherent and correct.

The authors present a methodology that allows analyzing diverse human realities integrating natural, social and economic factors. The methodology created is complex but well supported. The literature review is sufficient and current, supporting the research needs.

The created tool can be a great tool for urban, touristic and regional planning.

It would be interesting if the authors left open other readings of reality where the analysis of content and intangible aspects are out of the mathematical instruments.

Reviewer 3 Report

The paper focuses on the analysis of the spatial distribution characteristics and driving factors of traditional villages, which is very meaningful and interesting.  Unfortunately, I did not find any innovation in the methodology  or obtained very meaningful findings. The writing of the paper also needs to be improved. 

(1)In part of Independent variables ,the authors  mentioned Geographical factors.While, in the abstract and 4.3.2 GWR Model, the factor is regional culture factors. The same of different one?

(2) In the part of Methodology, it is mentioned Kernel density estimation method. While in the Results, the authors use Nuclear density analysis. It is better the same word if refer to the same method.

(3) In the conclusions, it is advised to focus more on providing new ideas for the protection of traditional villages, but not the analysis results.

Reviewer 4 Report

The author need to improve the title of the paper for example (

Spatial Distribution Characteristics and Driving Factors of Traditional Villages in the Awa Mountain Area Based on GWR 3
Modeling and a Geo-detector.

here the word distribution should be replaced by "distributed" or spatially distributed. Secondly, the word geo-detector should be replaced with other word.

 The author have to add more Reference to strengthen the paper.